# Non-Destructive Banana Ripeness Detection Using Shallow and Deep Learning: A Systematic Review

**DOI:** 10.3390/s23020738

**Published:** 2023-01-09

**Authors:** Preety Baglat, Ahatsham Hayat, Fábio Mendonça, Ankit Gupta, Sheikh Shanawaz Mostafa, Fernando Morgado-Dias

**Affiliations:** 1University of Madeira, 9000-082 Funchal, Portugal; 2Interactive Technologies Institute (ITI/LARSyS and ARDITI), 9020-105 Funchal, Portugal

**Keywords:** banana, computer imaging, deep learning, machine learning, ripeness

## Abstract

The ripeness of bananas is the most significant factor affecting nutrient composition and demand. Conventionally, cutting and ripeness analysis requires expert knowledge and substantial human intervention, and different studies have been conducted to automate and substantially reduce human effort. Using the Preferred Reporting Items for the Systematic Reviews approach, 1548 studies were extracted from journals and conferences, using different research databases, and 35 were included in the final review for key parameters. These studies suggest the dominance of banana fingers as input data, a sensor camera as the preferred capturing device, and appropriate features, such as color, that can provide better detection. Among six stages of ripeness, the studies employing the four mentioned stages performed better in terms of accuracy and coefficient of determination value. Among all the works for detecting ripeness stages prediction, convolutional neural networks were found to perform sufficiently well with large datasets, whereas conventional artificial neural networks and support vector machines attained better performance for sensor-related data. However, insufficient information on the dataset and capturing device, limited data availability, and exploitation of data augmentation techniques are limitations in existing studies. Thus, effectively addressing these shortcomings and close collaboration with experts to predict the ripeness stages should be pursued.

## 1. Introduction

Bananas are one of the most consumed foods in the world and the fourth most important food crop in terms of gross value of production, apart from wheat, rice, and maize. According to the Food and Agriculture Organization (FAO), they contribute about 16% of global fruit production [1]. Due to its rich nutritional value, they are extensively consumed, with almost 90% of manufacturing occurring in or around Asia, Latin America, and Africa [2]. This results in their high production demand worldwide, which ultimately depends on their post-harvesting phase. This phase constitutes the ripeness process, which is critical for deciding its nutritional quality. This process includes seven stages that can be identified by monitoring the fruit’s texture, chemical components, and their peel color [3].

The stage identification techniques are broadly categorized as destructive and non-destructive. The former is considered a reliable approach requiring the testing of bananas using peeling or cutting, whereas the latter does not require manual destruction. Specifically, the destructive approach deals with measuring dielectric parameters such as capacity, dielectric constants, resistance, and admittance changes throughout the ripening process of a banana [4,5]. The limitations associated with the destructive techniques are that they are expensive and require skill in identifying ripeness stages. On the other hand, the non-destructive approach does not require manual destruction. It examines the physical and spectral properties, such as color, texture, and chemical compositions, using “conventional machine learning and deep learning” approaches [6,7]. Its limitation is that it is relatively less reliable than the former, although it has received considerable attention in the recent decade [8].

Following the non-destructive approach, several studies have been conducted to predict the banana ripeness stages using imagery data [5,6,9] or sensor data [9,10,11]. Among them, studies employing the images were prevalent, utilizing state-of-the-art computer vision (machine or deep learning-based) algorithms. Several attempts have been made to compile state-of-the-art methods for banana ripeness stages. For instance, Dewi et al. [12] extensively reviewed machine learning methods emphasizing imagery data. The study demonstrated the potential of computer vision methods for the prediction of banana ripeness stages by extracting color and texture features of the images depicting banana ripeness. However, the review of Dewi et al. [12] did not include methods using sensor-based data, which disregarded the usability of nutritional composition changes during the progression of ripeness stages. Additionally, the review is limited to machine learning methods only. Another study by Muthu Lakshmi and Ranjith [13] presented deep and machine learning techniques for multiple fruits. Both review studies only emphasized imagery data, despite the importance of sensor-based data, which provide valuable insights. Furthermore, none of the reviews provided insights about designing the standardized study based on the factors affecting the performance of deep or machine learning methods.

This systematic review overcomes the above-mentioned limitations by summarizing both image and sensor-based studies. Additionally, this review also identified a set of critical factors that was used for categorizing the existing studies. The contribution of this systematic review is as follows:In the case of a banana ripeness review, this is the first work to remove selection bias using Preferred Reporting Items for Systematic Reviews and Meta-Analyses (PRISMA) guidelines [14] for the previous 20 years of published works and four well-known indexing services.Different fruits have different colors, textures, aromas, and maturity levels for consumption. Therefore, mixing different fruits may change the outcome of the review. Consequently, this review focuses solely on banana ripeness, unlike Muthulakshmi and Renjith’s [13] study, which has also included other fruits (mango, banana, jack fruit, durian, papaya, lemon, tomato, etc).Although for humans visual, touch, and smell may be the primary senses to detect ripeness, for automation, different sensors, including cameras, can be used. However, due to the dominance of camera uses, a comparison between various sensors has not been made before. This review tries to fill that gap by including different sensors along with cameras [12].A statistical analysis, as well as quality assessment studies, was performed to examine the model performance.Finally, the review objectively reports on datasets, banana features, ripeness stages, data augmentation techniques, and on the shortcomings of the current state of the art.

In this systematic review study, Section 2 and Section 3 summarize the eligibility criteria of the articles, information on vital parameters that consist of a detailed description and a quality assessment studies analysis. Section 4 presents the discussion that includes the context of evidence, the limitation of the studies, the recommendation and the future research part. Lastly, Section 5 presents the conclusion of the study.

## 2. Methods

This systematic review followed the new PRISMA guidelines [14]. This section summarizes the process of article search and analysis of the studies based on eligibility criteria for inclusion and exclusion of the study, information about the data sources, the data collection, and the selection process.

### 2.1. Inclusion and Exclusion Criteria

The experimental studies on banana ripeness, based on feature extraction and classification of banana ripeness stages, have been included. However, the exclusion criteria are based on the full articles that are unaccessible for which the relevant author was contacted but did not reply and due to the articles where the English translation was unavailable. Articles that focus on other fruit’s ripeness, banana disease, grading studies as well as the articles that do not focus on banana ripeness stages were also excluded.

### 2.2. Information Sources and Search Strategy

The database selection covered Science Direct, Scopus, Institute of Electrical and Electronics Engineers (IEEE), and Springer. The final database search was completed on January 15th, 2022. The process of screening the articles based on keywords was followed for this review, and the employed words for the search in each database are presented in Table 1. The article search was conducted between the year 2002 and December 2021. In this review, 20 years of work were summarized, especially those studies that used novel methods for detecting banana ripeness stages using any imaging modality were considered. All the references were managed, and duplicate hits were removed using Mendeley Reference Manager.

### 2.3. Screening Results Studies

In the search strategy shown in Table 1, a total of 1548 articles were retrieved using the four databases mentioned previously. Only 32 articles were finally included in the analysis. While screening these articles, four more studies were included for the final analysis by snowballing (verifying the list of every included article’s reference). The proposed work flow diagram, in Figure 1, follows the PRISMA principle to illustrate the article’s screening process. Appropriately, 35 articles were included for the final analysis. However, for the quality assessment studies, five studies have been excluded because no numerical results were presented in them. Furthermore, the investigation has not considered in vitro studies, since this review is limited to non-destructive methods for detecting banana ripeness stages. One article produced by Chmaj, Sharma and Selvaraj [15] could not be accessed due to the subscription for which the authors were contacted, but none of them replied. Another article by Li et al. [16] could not be included because the paper was published in Japanese and there was no English reference available for this paper. In recent years, the field of agriculture using Artificial Intelligence (AI) has been a challenging and hot research area. Therefore, it is essential to have a record of region-wise publications based on author affiliation with the number of studies presented on the basis of the country name. Most of the author’s studies are from Asian continents, and the details of the region-wise publications studies are shown in the Appendix A.

### 2.4. Data Collection and Selection Process

The data collected for the study were gathered from reading the publisher’s full articles and journals, which were grouped by country and year of publication. The selection process was based on the banana ripeness stages using AI. In the two-phase selection process, first, the researchers Preety Baglat (PB) and Ahatsham Hayat (AH) independently performed the full-text screening, based on titles and abstracts, for the selected articles. In the second phase, four other authors, Ankit Gupta (AG), Fábio Mendonça (FM), Sheikh Shanawaz Mostafa (SSM) and Fernando Morgado Dias (FMD), applied the selection criteria to the full text of the selected articles. Finally, the final discussion for the selection of the articles was made by all the authors.

### 2.5. Organized Data and Performance Evaluation

For all the studies, the data were collected after scanning the complete articles. Therefore, PB collected the study by checking these parameters such as types of banana samples, varieties of banana categories, wavelength or spectrum, measurement tool camera or sensor, color space, features, classification stages, number of data samples, methods and results.

The primary outcome of this review is the comparison of various banana ripeness detection methods using multiple imaging techniques with the respective capturing devices. The secondary outcome of this study is the evaluation of the performance of all the included studies by their relevant parameters and the analysis of practical significance and limitations by the performance metrics. Moreover, a similar performance metrics report was selected for all the included studies, and the median and mode were used for the study categorization.

### 2.6. Study Quality Assessment

This review aims to find the available research on banana ripeness stages’ detection using machine learning methodologies. The quality assessment was categorized based on scores and is presented in the Appendix A. The assigned scores are based on the following parameters: banana capturing devices, the types of banana stages, the total number of samples in the dataset and the reported performance (based on accuracy and the coefficient of determination values). For the quality assessment studies, a total of 31 (from the 35) studies have been included for further analysis, as five studies have been excluded because no numerical result was presented in them. The included studies have been divided into three categories, namely, a non-augmented dataset (composed of 22 articles), a highly augmented dataset (with four works), and a sensor-based dataset (containing five studies). Each study has been assessed based on four parameters: the capturing devices, the number of samples, the types of banana stages, and the reported results (accuracy and the coefficient of determination). All parameters were assessed based on specific criteria and were awarded a score of either 1 or 0.

For the first parameter (capturing devices), a score of 1 was awarded if detailed information about the device was provided; otherwise, a score of 0 was given. Similarly, for the second parameter (the total number of samples), the median of the quantitative value was calculated by combining the score of individual values from each study. A score of 1 was assigned if the total number of samples was greater than the median value; otherwise, the value 0 was assigned. In the following parameter (types of bananas ripening stages), the mode of the number of ripening stages was calculated by combining the score of individual values from each study. If the number of stages was greater than 3 (mode value), then a score of 1 was given; otherwise, the score value was 0. For the last parameter (results), the median of the quantitative value was calculated by combining the score of individual values from each study. A score of 1 was awarded if the result was greater than the median value and it was 0 otherwise. After this analysis, all individual score parameters were summed up for each study to attain the overall quality assessment.

Finally, the summed-up score was categorized into three types: strong, fair and weak. A study was considered strong if the overall score was 4. A fair study occurred when the overall score was 3; lastly, a study was marked weak when the score was 2 or less.

### 2.7. Result Tabulation and Visual Interpretation

In this review, Table 1 and table in Section 3.3 below depict the mean and standard deviations (SD) values of the examined metrics, details of the results (accuracy, value) and assessment scoring. In the Appendix A were used for identifying the essential parameter’s detailed information, the overall performance metrics based on the accuracy, the coefficient of determination values of the studies and detailed information on the scheme that has been used for the quality assessment scoring in this systematic review. In the Appendix A, bar plots and a map chart in Appendix A were used to visualize the grouping of the banana samples and the types of banana stages. Appendix A depicted a region-wise publication based on the author’s affiliation for the detection of banana ripeness.

### 2.8. Missing Data, Subgroup Analysis and Heterogeneity

The proposed study penalizes missing information, which is helpful for the quality assessment study. This review provides a descriptive summary based on data collected from all the included studies for detecting banana ripeness stages. All the authors have gathered the data from individual studies and performed the quality assessment based on the collected data, yet the subgroup analysis was not performed. To support the effectiveness of the models, the proposed protocol calculates the median of the performance metrics.

## 3. Results

This section explains banana characteristics, which include the description of the grouping of the banana samples and the varieties of banana categories. Additionally, crucial parameters, such as capturing devices (camera or sensor), datasets, the different stages of banana ripeness, feature analysis, classification, and the feature-based methods and performance metrics, are useful for detecting and classifying banana ripeness stages.

### 3.1. Banana Characteristics Based on Grouping of Samples and Variety

The terminology for banana analysis includes terms such as bunch, fingers, and hand. A single plant produces bunches of bananas. A single banana is called a finger, and a grouping of attached fingers of 10 or more bananas is called a hand [3]. In this review, out of the 35 studies, a total of 27 [3,4,5,6,9,10,11,17,18,19,20,21,22,23,24,25,26,27,28,29,30,31,32,33,34,35,36] have used the fingers of bananas, 3 [37,38,39] studies have used the fingers by hand and Chen et al. [40] used a bunch of bananas. Mohamedon et al. and Rodrigues et al. [41,42] used a mixed sample of bananas, Mendoza and Aguilera [43] used the combination of fingers by hand and a single batch of bananas, and whereas Saragih and Emanuel [44] also used the used the combination of fingers and fingers and hand bananas. The grouping of banana sample collection is mentioned in the Appendix A.

For the detection of banana ripeness stages, most of the studies have used Musa Cavendish varieties such as the Musa cavendish (Musa spp, Musa Paradisiaca, Musa AAA (Cavendish Subgroup) group (Japan), the Musa Cavendish Ambon banana, Red, Musa basjoo Siebold var. Formosana, Musa acuminate, Musa Acuminata cv. Berangan, Embun, Mas and Berangan, Musa AAA group (Kluai Hom thong), the Musa AAA group (Brazil), and Egyptian species. Out of the 35 studies, most of the studies (21) have used Musa cavendish banana categories. Apart from that, three [6,9,25] studies used the Egyptian species, and Sabilla et al. [21] used local bananas from traditional markets in Indonesia (Pisang Emas, Pisang Kepok, Pisang Ambon, Pisang Raja Nangka, Pisang Santen, Pisang Susu, and Pisang Candi). Likewise, Saragih and Emanuel’s [44] study used the combination of two categories (Egyptian species and an unreported category), while the remaining 10 [11,18,24,26,30,34,37,38,41,42] studies did not report their banana categories.

### 3.2. Study Design

In this section, the detection of banana ripeness stages using the ML pipeline is displayed in Figure 2. All the different parameters, such as the grouping of banana samples, the types of capturing devices, the types of banana stages, a list of banana features, classification and the feature-based methods, the performance metrics, the challenges, and a quality assessment analysis of existing studies used for the detection of banana ripeness stages, are discussed in detail below.

#### 3.2.1. Capturing Devices and Techniques

The non-destructive studies require camera or sensor-based devices for image acquisition. Hence, camera or sensor characteristics such as the camera type, the wavelength distance, and the sensor type are effective parameters for banana ripeness identification. Therefore, this section summarizes the relevant articles’ sensor types, the wavelength distance, and the color space distribution for the included studies. Out of the total considered studies (35), 18 [5,6,9,10,19,20,21,24,26,27,28,34,38,39,40,41,42,44] used a standalone digital camera. Hou et al. [22] used spectrometers and Saadl et al. [30] used a webcam. Furthermore, Xie, Chu and He [33] used hyperspectral capturing devices and Chen et al. [40] used a combination of a digital camera, an Electronic-nose (E-nose) using a Data Acquisition Card (DAQ), and an E-nose or camera.

A total of four of the 35 [4,17,18,35] studies have used the combination of Laser Light Backscattering Imaging (LLBI) with a wavelength distance and a Charge-Coupled Device (CCD). Another three [3,23,43] studies have used fluorescent lamps and digital cameras. A combination of a smartphone with visible and fluorescent spectral images was used by same authors [29,32]. Chen et al. [40] used a combination of a digital camera, an Electronic-nose (E-nose) using a DAQ, and an E-nose or camera. Lastly, Mueez [25] used the combination of a Raspberry Pi 4 model B with a sonar filter to the back of the wall of a structure carved out of cardboard and a camera.

While the rest of the studies (five of the studied 35) have used sensor-based devices, Altaf et al. [37] used a set of wireless sensors for measuring the temperature, the humidity level, the ethylene concentration, and the carbon dioxide level. Suthagar et al. [11] used the combination of an electronic nose employing the MQ3 (ethylene gas) sensor detecting the gases, the MQ6 sensor (propane, butane, and olefins), the MQ8 sensor (which is suited for detecting the gases of hydrogen), the MQ135 sensor (air quality control: ammonia, benzene, hydrogen), an Arduino board, an LCD, and an array of sensors and camera images. The combination of the MQ3 (ethylene gas) sensor, digital temperature and humidity sensor (DHT22), and an Arduino Uno with Graphical User Interface (GUI) was used in Taghoy and Villaverde’s [31] work. Lastly, Sanaeifar et al. [36] employed an electronic nose using the MQ3 (alcohol), the MQ135 (air quality control), the MQ131 sensor detecting the ozone, the MQ136 (sulfurated hydrogen), and the MQ5 sensor (which is suited for detecting the LPG, natural gas and coal gas). Chen et al. [40] used a combination of a digital camera, Electronic-nose (E-nose) using a DAQ, and an E-nose or camera using MOS gas sensors which includes TGS2600 (hydrogen, carbon monoxide), TGS2602 (ammonia, hydrogen sulfide), TGS2603 (trimethylamine, methyl mercaptan), TGS2610 (butane, LPG), TGS2611 (methane, natural gas), TGS2612 (methane, propane, iso-butane) and TGS2620 (alcohol, solvent vapors). The types of banana-capturing devices are summarized in Figure 3.

#### 3.2.2. Dataset Description and Statistics

For the banana ripeness analysis, 28 of the 35 studies used a self-created dataset to prove the significance of their proposed techniques. Additionally, the remaining eight [9,24,25,26,38,39,41,44] studies have used a publicly available dataset. Out of the 35 studies, only five [5,10,36,37] studies have provided the origin country information of the dataset, of which two [10,36] studies came from the Philippines, Altaf et al. [37] were in Pakistan, and Mohapatra, Shanmugasundaram and Malmathanraj’s [5] study is in India. The rest of the studies have taken the data from commercial distributors, local markets, and self-created datasets but did not provide the origin country information, using existing online resources data and data from the ripening facility in Potsdam Germany, whereas 12 [11,19,21,27,28,29,30,31,34,38,39,41] studies did not report any information about the country origin of the dataset.

Medians were calculated from all the included augmentation and sensor studies for the quality assessment analysis. Out of all 30 quality assessment studies, 12 [6,9,10,17,18,19,20,25,35,38,41,44] have used 245 or more samples and 11 [4,5,22,23,27,30,33,37,40,43] studies have used values less than this amount. Among the four articles that have used a highly augmented dataset, Sabilla et al. [21], as well as Marimuthu, Mohamed and Mansoor Roomi [39] have employed a median value of 2054 or more samples, and the other studies of Spaches [26] and Vaviya et al. [34] used values less than this. For the four studies that have used a sensor dataset, Suthagar et al. [11] and Sanaeifar et al. [36] used a median value of 11.5 or more samples, whereas Altaf et al. [37] and Taghoy and Villaverde [31] used fewer than this.

#### 3.2.3. Stages of Banana Ripeness

The banana texture alterations, starch breakdown, unexpected fragrance, pigment creation and abscission are all brought on by a series of biochemical changes that occur during the ripening process. The color of a banana peel changes as it ripens, going from green to yellow to brownish. The most common criterion by which consumers judge the actual and consumable quality of a banana is its peel color. Bananas lose some of their hardness when they ripen, which is also a sign of quality [37]. Therefore, it is important to identify the right stage of banana ripeness. In this section, a discussion of the different banana stages used by existing studies is presented: three stages (unripe, mid-ripe, over-ripe); seven stages (green, green traces of yellow, more green than yellow, more yellow than green, green tip, yellow, and yellow flecked with brown); four stages (unripe, yellow-green, mid-ripe, and over-ripe); two stages (ripe and unripe); six stages (natural green, light green with light yellow, yellow with light green, yellow with brown spots, full yellow, and yellow with green ends) and 14 stages (the type of stages are not mentioned in this particular study). The types of banana stages are mentioned in the Appendix A.

There are different stages used for the classification of banana ripeness. Among the 35 examined articles, 4 [5,22,36,43] examined seven stages, 4 [17,18,35,37] studies have used six stages, 6 [6,19,20,25,40,44] have used four stages, 12 [9,10,11,24,27,29,30,31,32,39,41] used three stages, and the remaining 3 [33,34,41] studies have used two stages. Lastly, Sabilla et al. [21] used the seven categories of banana, and Iswari et al. [28] employed three different types of fruits, namely, apple, banana and melon.

The remaining articles used a multi-stage analysis. Particularly, Zhu et al. [26] used three stages and two stages, while Zhang et al. [23] used seven and 14 stages. Likewise, Zulkifli et al. [4] used six and two stages. One of the 35 [3] studies has not reported the number of ripeness stages used. For the quality assessment analysis, modes were calculated from the included augmentation and sensors-based studies.

#### 3.2.4. Features Analysis

The banana ripeness stages identification can be conducted by external and internal characteristics, such as the color space (RGB (Red, Green, and Blue) and HSV (Hue, Saturation, and Value)), L*a*b, where L* values represent the brightness components of a banana that go from black (0) to white (100) values, a* (green to red) parameters represent the greenness and b* (blue to yellow) represent texture; SSC (Soluble Solid Content); a brown spot feature; chlorophyll; firmness; pH (potential of hydrogen); shape; the ratio of pulp to peel; light and ultra-white ration, etc. These attributes are the potential criteria for identifying various banana ripeness stages. Therefore, as the ripeness stages change from unripe, yellow-green, mid-ripe, and over-ripe, the hue color gradually changes from green to yellow; hence, it is essential to be undertaken. Taking the color space distribution into account, the hue of the banana ripens from fully green to yellow, and this serves as a crucial visual cue to distinguish between the various phases of ripeness [20]. However, most of the promising result studies [6,20,22,24,26,40,43,44] have used RGB, HSV, L*a*b* and texture feature for the development of efficient and non-destructive methods that can easily be implemented by using computer vision techniques.

Although all RGB color space features simultaneously contain brightness and color information, the brightness information may not be useful for ripeness stage identification. Due to the independence of the color components in these spaces from the equivalent brightness components [20,26], the HSV and L*a*b* color space features proved to be useful in providing pure color information.

Another inherent quality of a banana fruit surface is its texture. It describes how the gray area is distributed across the image area and the pixels, and it remains constant under changing lighting. The texture of a banana surface is the single aspect that can properly capture all its important characteristics. Nevertheless, obtaining high-level image representation using simply texture information is not yet achievable [26]. However, the SSC feature, which corresponds to the softening of the banana fruit tissue, rises as the banana fruit ripens. The water relations in the banana fruit tissues are modified because of changes in cell wall characteristics that take place during ripening, and this causes solute reorganization between tissue compartments. In contrast, the chlorophyll feature degrades the result of the banana-ripening process, resulting in the color of the banana peel changing throughout, most notably from green to yellow in the later stages [18]. Other characteristics, such as firmness, pH, and the ratio of pulp to peel, light and ultra-white ration, etc., have also been used to evaluate the quality and ripeness of bananas, and they are now recognized as significant indicators of ripeness, changes in quality indices of the banana at various ripeness stages, and their potential shelf life [10]. Therefore, it is important to identify the most crucial features used for the detection of banana ripeness stages.

Color space is essential for extracting accurate banana ripeness stages. Out of all 35 included studies, three [30,38,41] used color features, and Quevedo et al. [3] employed fractal texture features. Furthermore, Hou et al. [22] used the L*a*b* color space feature, whereas Sabilla et al. [21] and Mueez [25] have used peel color.

In this review, all the combinations of feature studies were included. A total of three out of 35 [17,18,35] studies have used the combination of chlorophyll, elasticity, and SSC features. Only Mendoza and Aguilera’s [43] study used the L*a*b*, brown spot percentage of total area (BSA), and number of brown spots per surface (NBS and texture features). In contrast, Mazen and Nashat [6] used color, BSA, and texture features.

In the remaining 26 studies, Zhuang et al. [20] utilized the combination of peel color, local texture and local shape (top, middle, and the tip) using a local binary pattern with uniform patterns (UP-LBP) and local shape features using a histogram of oriented (HOG). Meanwhile, Cho and Koseki [10] used the firmness, total soluble solids (TSS), pH, and the ratio of peel features. Zhang et al. [23] used the combination of the banana’s color, shape, and texture among with local and global features. Maimunah et al. [27] and Adebayo et al. [44] used the Red Green Blue (RGB), hue saturation value (HSVL), L*a*b*, and the texture of the banana as features. In contrast, Altaf et al. [37] used temperature, humidity level, ethylene concentration, and the carbon dioxide level features. Furthermore, two [28,36] articles used the combination of three colors, the number of dots (yellow, green, and brown) and the sweetness level.

Chen et al. [40] employed aroma and color features, while Zhu and Spachos [24,26] used HSV and texture features. Suthagar et al. [11] and Vaviya et al. [34] proposed the use of color and shape or size features. Furthermore, Marimuthu, Mohamed Mansoor Roomi [39] used the pH and NBA (Normalized Brown Area) features, while Zulkifli et al. [4] employed the L*a*b*, firmness, TSS, and pH features. In two articles, Intaravanne, Sumriddetchkajorn and Nukeaw [29,32] used a combination of white and ultraviolet light ratio features, whereas Taghoy and Villaverde [31] used the temperature and ethylene gas as features. Xie et al. [33] used the L*a*b and firmness feature, and another study of Mohapatra, Shanmugasundaram and Malmathanraj [5] employed the combination of capacitance, relativity, permittivity, impedance, and admittance changing during ripening process features. Finally, Ramadhan et al. [19], Saranya, Srinivasan and Kumar [9] and Rodrigues et al. [42] have not reported their used features.

#### 3.2.5. Classification Methods

*Machine learning-based methods*: Out of the 35 included articles, Zulkifli et al. [4] used the Linear Discriminant Analysis (LDA) method, while five [10,17,18,30,37] used an ANN method and two [22,35] employed an SVM model. The remaining two [25,28] studies used the KNN model; Mendoza and Aguilera’s [43] study used the Sequential Forward Selection (SFS) feature selection method and discriminant analysis (DA) method for classification, and Quevedo et al. [3] employed the fractal Fourier method. Lastly, Xie, Chu, and He [33] used Partial Least Squares Discrimination Analysis (PLS-DA), Kipli et al. [38] used the J48 classification method, and two more articles [31,39] used Fuzzy Modeling for the Classification of Banana Ripeness (FMCBR).

Choosing suitable methods for the classification and the feature creation for banana ripeness are essential in identifying the ripeness stages. Figure 4 depicts the distribution of all the identification methods used for banana ripeness on the basis of the number of times the methods appeared in the studies. Different types of classification procedures were used: namely, manual feature-based classifiers, automated feature extraction and classification methods, feature selection procedures and the list of features used by all included studies for the detection of banana ripeness stages. In automated feature extraction and classification methods, four studies have used a normal convolutional neural network (CNN) method, and the other studies have used well-defined state-of-the-art CNN-based architectures. Local receptive fields, weight sharing, and spatial pooling layers are three major characteristics that distinguish CNN from a generic neural network. CNN is typically a multilayer, hierarchical neural network. By forcing each neuron to solely rely on a spatially small subset of the neurons in the preceding layer, CNN uses a local receptive field as opposed to a global one, mimicking how the brain captures local structure in images. In CNN, spatial pooling is accomplished by first breaking the image up into an array of blocks, after which a pooling function is assessed over the responses in each block. Pooling is intended to minimize the dimensionality of the convolutional responses and enforce (to a limited extent) translational invariance in the model. The response for each block is assumed to be the maximum value over all response values within the block in the case of max pooling. Multiple layers of a conventional CNN alternate between convolution and pooling. Deep CNN contains more hidden layers than shallow CNN systems. Low-level convolutional filters, which can be regarded as giving low-level encoding of the input image, are built from lower layers, which are defined as those closest to the input. Deeper layers, on the other hand, pick up more intricate structures. The number of pixels by which the local receptive field is pushed to the right in CNN is determined by the stride length. CNN architecture categorizes the bananas at various phases of ripening and forecasts the likelihood of classification of the input image [23]. For using deeper CNN models (such as in the Residual Neural Network), the bottleneck-based archetecture was developed allowing to overcome the gradient-related problems associated with deep CNN [45]. In terms of manual feature-based classifiers, white box and other types of models are considered in almost 90% of the studies. Most of the studies have used color, L*a*b* and texture features, in terms of classification-based methods support vector machine (SVM), K-Nearest Neighbors (KNN), ANN, LDA, and CNN, whose models perform best when compared with other classification-based methods.

Mazen and Nashat [6] used a combination of ANN, SVM, Naïve Bayes, KNN, and decision tree methods. Likewise, Zhuang et al. [20] used Naïve Bayes, LDA, and SVM-based models, while Sabilla et al. [21] employed KNN, SVM, and decision tree methods. Simultaneously, Maimunah et al. [27] used Naïve Bayes and ANN classifiers.

*Deep learning-based methods*: Furthermore, four [19,23,34,41] articles have used a CNN method. Out of all the combination methods used in the studies, three of the examined 35 [9,42,44] studies have used CNN-based models, namely, VGGNet16 and ResNet50 using transfer learning, VGG (detection and classification) and VGG-16 (object detection; MobileNetV2 and NASNetMobile (pretrained model)). In another work, Suthagar et al. [11] employed the E-nose, image processing, and manual techniques. The remaining two [29,32] studies performed manual analysis while not analyzing the results.

*Machine and deep learning-based methods*: Only Zhu and Spachos [24] used the combination of two classifiers, using KNN, Random Forest (RF), Naïve Bayes (NB), and an SVM model as classifier I, and a YOLOv3 (You Only Look Once Version3) model using transfer learning as classifier II. Similarly, Zhu and Spachos [26] used a combination of an SVM and the YOLOv3 models, while Sanaeifar et al. [36] used principal component analysis (PCA), Soft Independent Modelling of Class Analogies (SIMCA), LDA and SVM models, and another Mohapatra, Shanmugasundaram and Malmathanraj [5] article used KNN and Fuzzy C-Means (FCM) methods.

### 3.3. Performance Metrics

The performance analysis of the banana ripeness identification stages is based on two metrics, namely, accuracy and $R^{2}$. In these two metrics, most studies used accuracy, and only three studies have used coefficients of determination. However, out of a total of five graphical results studies [3,28,29,32,42], one article used the graphical representation with a fractal dimension from 2.02 to 2.20. Two studies have provided the probabilistic object identification-based output and application-based output (no numeric value), while the remaining two studies have provided the image-based output. The overall performance metrics of accuracy and the coefficient of determination are mentioned in the Appendix A. Mean and SD values of the examined metrics and the details of the results (accuracy, value) are shown in Table 2 and Table 3 below. Medians were calculated from all the included augmentation and sensor studies for the quality assessment analysis. Out of all the 23 non-augmented studies, 12 studies have attained an accuracy (median value) of 95.37% with a coefficient of determination median value of 0.82 or more, while 11 studies have reached values lower than this. In contrast, four studies have used a highly augmented dataset (two out of the four studies have reached an accuracy (median value) of 96.7% or more, and two studies had values lower than this), four studies have used a sensor dataset (two out of the four studies have used accuracy (median value) of 95.26% or more, and two studies used values less than this).

### 3.4. Challenges

Different types of camera or sensor devices are used for the analysis and detection of banana ripeness. LLBI device studies [4,17,18,35] required special sensors and lighting; nonetheless, all the cameras were not integrated with these sensors. Hyperspectral and multispectral images have many channels that can be used to detect a specific character. This type of device is not only resilient for remote sensing applications but also proves to be helpful for the identification of banana ripeness stages, which can be shown by high recognition accuracy. However, expensive equipment is necessary for the acquisition of proper hyperspectral imagery.

The sensor devices measure the temperature, the number of dots on a banana skin, the sweetness level, the humidity level, the ethylene concentration and the carbon dioxide level. However, sensor-based devices are not widely found for banana ripeness detection. The sensor devices have a more complex installation, and low-cost sensors do not provide information on the exact position. Secondly, most of the included studies have used a small number of data samples, formally decreasing the classification accuracy. Whereas other [21,26,34,39] studies have used highly augmented data samples, which can be used to build a more accurate model, but occasionally, this alleviates the overfitting issue when large deep neural networks are trained. However, there are many encouraging results, yet it is not guaranteed that the data augmentation can improve the generalization error. Training with highly augmented data will lead to a smaller robust error but also a potentially larger standard error [46]. These critical challenges need to be addressed for the detection of banana ripeness.

### 3.5. Quality Assessment Studies Analysis

Regarding the quality assessment analysis, four parameters were considered for identifying banana ripeness studies: namely, the capturing devices, the dataset, banana ripeness stages, and the results (accuracy and $R^{2}$). Based on four parameters, details of the quality assessment studies are presented in Table 4. All included studies (non-augmented dataset-based, highly augmented, and sensor-based dataset studies) were categorized into three types: namely, weak, fair, and strong. Based on the proposed quality assessment analysis protocol, out of the 22 (from the 35) actual dataset-based studies, 6 [4,23,24,27,33] were identified as “weak”, 15 [5,9,10,18,19,20,22,25,30,35,38,40,41,43,44] were recognized as “fair”, and 2 [6,26] were classified as “strong”. On the other hand, out of the four of the 35 articles that composed the highly augmented dataset-based category, Zhu and Spachos [26] were identified as “weak”, two [34,39] were identified as “fair”, and Sabilla et al. [21] were found to be “strong”. Regarding the sensor-based studies (five of the examined 35 articles), three [11,26,31] were identified as “weak”, that of Altaf et al. [37] was found to be “fair”, and that of Sanaeifar et al. [36] was identified as “strong”. The remaining five [3,28,29,32,42] of the 35 studies are not included in the quality assessment because they reported the results graphically. Detailed information on the following scoring, a scheme that has been used for the quality assessment in this systematic review, is shown in the Appendix A.

## 4. Discussion

This review summarized the existing non-destructive techniques and works for the detection of banana ripeness. Identifying the proper banana ripeness stages is essential for banana storage, transportation, and quality control marketability [19]. Therefore, this review’s comparisons and analysis of multiple factors in included studies would be helpful for the researchers to select the useful parameters wisely, such as features, accuracy, the coefficient of determination, and classification models for identifying banana ripeness stages.

### 4.1. Context of Evidence and Limitation

Most studies have used the Musa Cavendish variety of bananas for banana ripeness detection because it is the most frequently produced for export markets and publicly available in the dataset. Therefore, this composes a limitation of these studies, as few have used other banana species. Selecting the correct banana-capturing device is crucial for detecting banana ripeness. The capturing devices used in most studies are camera-based, making it a cost-efficient solution for the real-time banana monitoring system. However, detecting certain important features of the banana development stages is impossible for the camera-based device. At the same time, few studies have used a visible fluorescent spectral images device with a wavelength of 365 nm and a hyperspectral imaging (HIS) device, which are powerful imaging techniques used to analyze banana quality. In one of the included studies, HIS was used to predict banana firmness, color space features, and the ripeness stages using the 380-1023 nm wavelength. The model’s performance was examined by the overall coefficient of determination of 0.795, and L*a*b* firmness and its residual predictive deviation values were 2.119, 2.234, 2.062, and 6.098, respectively [35]. This technique is faster and more effective for scanning more samples simultaneously.

Few studies have used a laser light backscattering imaging device based on a monochromatic laser light source. LLBI comprises a series of monochromatic images captured at different wavelengths (532–1060 nm). When a monochromatic laser light source reaches the sample surface, some photons are reflected, while the rest of the light beam is scattered. Light absorption depends on chemical constituents such as chlorophyll, elasticity, and SSC. Scattering, which is a natural phenomenon, occurs due to the microstructural composition and cellular matrices. Therefore, LLBI is a preferable monitoring technique for banana ripening detection. These capturing devices displayed the potential to evaluate the ripening stages of bananas quickly and accurately without interrupting the composition. The included studies suggest that these innovative approaches can overcome the cumbersome, destructive, complex, and time-consuming sampling associated with traditional analytical tools. However, the limitation of these devices is their high cost and the need for more expertise for the experimental setup, which requires a high processing time to extract the important attributes from the spectral images. The condition and stability of the thermal behavior of the agricultural products, under various environmental factors, determine the accuracy of their results. Moreover, integrating these technologies with other sensing devices (wireless sensor, E-nose using MQ3, MQ6, MQ8, MQ135, MQ5, MQ8, MQ131, MQ136) may pave the way for a more accurate detection for the banana ripeness stages. Yet, the problem with these sensing devices is that they require a proper experimental setup with expertise.

Furthermore, in most of the included studies, with relation to the other machine learning methods, ANN performed better for the classification and LDA performed better for the feature creation method (of the banana ripeness stages) when the dataset is small. When the dataset is large, it was observed that CNN models performed better. Regarding the sensor-based study, the SVM model was the most relevant. In terms of white box-based model performance, the Decision Tree model [6] has performed well when compared with other methods such as ANN, SVM, KNN and Naive Bayes. As there are very limited studies conducted using a white box-based model, therefore, in the future, tree-based algorithms can be explored more. Existing approaches use assumptions to process the data, which is not preferred by deep neural networks. The main limitation with most of the studies is that the dataset was too small to properly develop a CNN-based model, leading the researchers to use highly augmented data, which can lead to the overfitting problem for the classifier.

For the model to perform better, identifying the correct number of banana ripeness stages is crucial. Most included studies have used three stages (unripe, mid-ripe, and over-ripe) to classify banana ripeness. However, only limited studies have used the standard seven stages of banana ripeness, which has helped the model classify the ripeness stages more accurately. In addition, there has not been any study that includes an independent dataset, and such a situation can arise in a real-world scenario when the dataset is completely new to the system [47], such as the introduction of a red banana species to the classifier. To classify the unlabeled dataset, an unsupervised learning mechanism can be implemented to train the model: for example, K-Means clustering can help in image segmentation of the unlabeled input dataset. Lastly, this review discovered common factors applicable to the non-destructive approaches to identify the banana ripeness stages parameters. Capturing devices, datasets, the classification of different banana stages, and performance metrics, depicted by accuracy and the coefficient of determination, are the common factors used by all the included studies. In general, the interest of researchers can be attracted toward the advancement in the computational speed of algorithms and data analysis, improvements in image processing approaches for real-time applications, and the development of low-cost imaging equipment.

### 4.2. Limitations of the Review Process

There are challenges and limitations acquired while compiling this review. Out of all the included studies, five [3,28,29,32,42] represented visual results only; hence, it became unfeasible to extract the numerical values for the quality assessment analysis properly. Additionally, the search strategy aimed at this review was to summarize and analyze the new methodologies for detecting banana ripeness stages. Therefore, this study excluded the other commercial applications in these areas. In addition, comparing the analysis of sensor-based devices, actual dataset-based studies, and highly augmented studies together proved to be challenging. Because of this, all the included studies were analyzed separately, which might lead to biased scoring. Due to the database search strategy, the lead-time and reporting bias of the collective research articles may be possible. As a result, several factors might affect the “risk of bias” quality assessment among all the included studies.

### 4.3. Recommendations and Future Research

This review identified the four key parameters for banana ripeness detection, which should be addressed while designing the review.

The sample collection parameter affects the efficiency of the model. Future works should validate their works on larger datasets using standard machine learning methods reported in a standard notation. There is a gap in deep learning-based solutions that should be addressed, using large datasets that should avoid using too much augmentation. More work should be carried out on banana bunch detection in real-world scenarios, as most works aim to address only the finger-based analysis in perfect laboratory conditions.In addition, classifying the correct number of banana ripeness using four stages can aid the researcher in using the correct stages for extracting maximum information for building an accurate banana ripeness model. In most of the best results, four ripeness stages were used with color features (RGB, HSV, L*a*b*) and the texture of the banana.The best performance was obtained using color-based classifiers, which was probably due to the significant differences in banana ripeness stages when compared to the other features. As a result, in the future, a low-cost method based on external features could be used to identify the automatic banana ripeness stages.It is important to identify the possibility of the mis-classification of banana ripeness stages by using other performance metrics such as confusion metrics. In our review study, most of the studies have used accuracy and $R^{2}$. In future work, the researcher can also consider other performance metrics such as sensitivity and specificity, which can be helpful for real-time scenarios.

## 5. Conclusions

This review study summarized the existing deep and machine learning techniques available for the identification of banana ripeness stages. Furthermore, the critical factors related to the data gathering, data pre-processing and classification such as capturing device/sensor, banana ripeness stages, appropriate features and classification methods are identified in this systematic review and can be used for designing a standardized study in the future for predicting accurate banana ripeness stages. The statistical analyses conducted under this review study can allow the researchers to compare their respective work with the performance metrics of existing techniques. It is visible from the existing studies that for the small dataset, ANN performed better for problems related to classification and LDA performed better for feature creation. Whereas, in the case of large datasets, CNN has performed significantly well. The performance of models varies with the change in data collecting devices/sensors: the SVM model works well when data are gathered by sensors instead of a camera. This review highlights the gaps in the existing banana ripeness stages identification techniques and the possible ways to fill those voids in future studies.

Finally, to enhance the quality of future work, we proposed in this systematic review the following recommendations: use of unlabeled datasets; avoid a large extent of data augmentation; consider color features and appropriate stages; use other performance metrics.

## Figures and Tables

**Figure 1 sensors-23-00738-f001:**
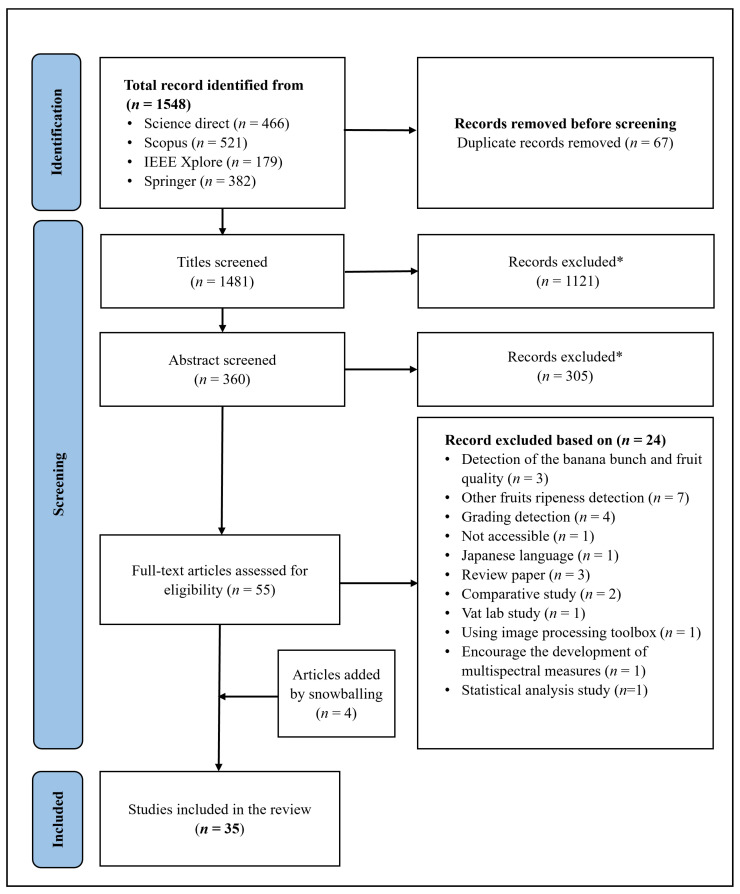
PRISMA flow diagram for banana ripeness.

**Figure 2 sensors-23-00738-f002:**
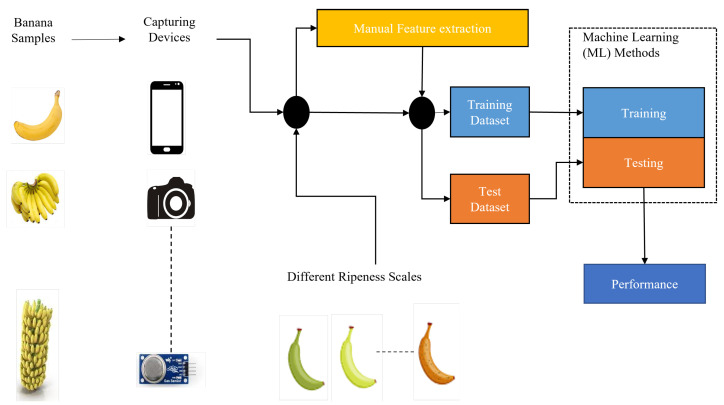
Banana ripeness detection using an ML pipeline.

**Figure 3 sensors-23-00738-f003:**
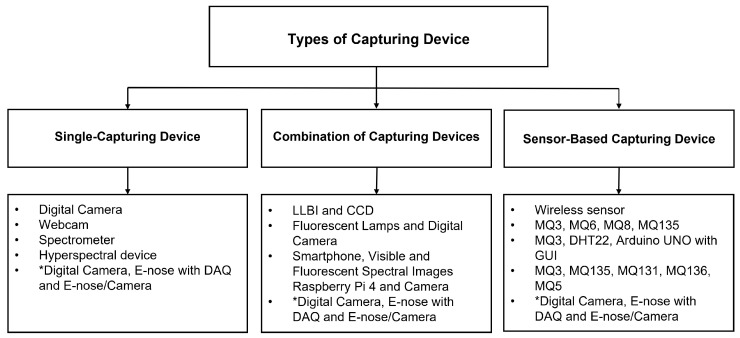
Types of capturing devices used for the detection of banana ripeness studies (* represented the study approaches in all three categories).

**Figure 4 sensors-23-00738-f004:**
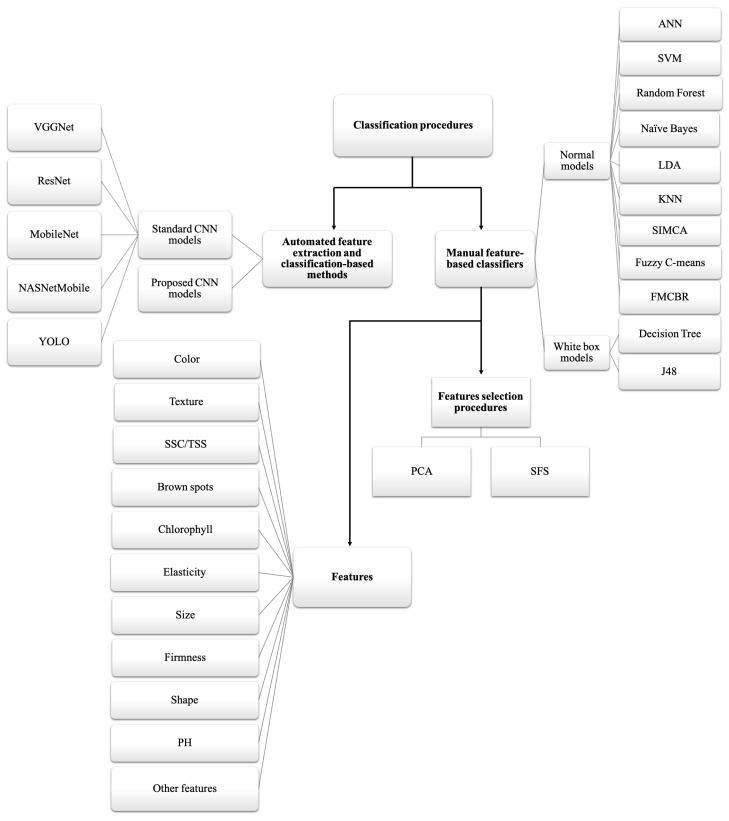
Classification and feature-based methods.

**Table 1 sensors-23-00738-t001:** Search strategy.

Data Resources	Search Field	Keywords
Science Direct	title	(Banana ripeness stages AND Deep learning classification) OR (Banana ripeness stages AND Machine learning) OR (Banana ripeness stages AND computer Imaging)
Scopus	title	
IEEE Xplore	title, full text and metadata	
Springer	title	

**Table 2 sensors-23-00738-t002:** Automated and manual feature-based classification accuracy for banana ripeness detection (* represented the study provided the overall manual and automated method accuracy).

Types of Studies	First Author [Reference], (Year)	Accuracy (%)
Manual	Mendoza [43], (2004)	98.0
	Zhuang [20], (2019)	91.2
	Mazen [6], (2019a)	96.9
	Adebayo [18], (2017)	95.5
	Sabilla [21], (2019)	96.6
	Suthagar [11], (2021	94.4
	Hou [22], (2015)	95.7
	Adebayo [35], (2017)	92.5
	Mohapatra [5], (2017)	96.1
	Zulkifli [4], (2019)	94.2
	Maimunah [27], (2019)	92.9
	Saadl [30], (2009)	89.0
	Kipli [38], (2018)	91.6
	Overall Mean ± SD	94.2 ± 2.6
Automated	Adebayo [17], (2016)	97.5
	Mueez [25], (2020)	85.0
	Ramadhan [19], (2020)	71.9
	Zhang [23], (2018)	93.5
	Mohamedon [41], (2021)	98.2
	Saragih [44], (2021)	93.5
	Overall Mean ± SD	89.9 ± 10
Manual (Highly Augmented Data)	Sabilla [21], (2021)	97.8
	Zhu [26], (2020)	98.5
	Zhu [24], (2021)	98.5
	Marimuthu [39], (2017)	93.1
	Overall Mean ± SD	96.9 ± 2.6
Automated (Highly Augmented Data)	Vaviya [34], (2019)	97.0
	Zhu [24], (2021)	85.7
	Zhu [26], (2020)	89.5
	Overall Mean ± SD	90.7 ± 5.7
Manual (Sensor-Based)	Sanaeifar [36], (2014)	97.7
	Chen [40], (2018)	92.6
	Altaf [37], (2020)	96.8
	Overall Mean ± SD	95.7 ± 2.7
Manual and Automated (Sensor-based) *	Suthagar [11], (2021)	94.44

**Table 3 sensors-23-00738-t003:** Coefficient of determination ($R^{2}$) for banana ripeness detection.

First Author [Reference], (Year)	Coefficient of Determination ($R^{2}$)
Cho [10], (2021)	0.79
Taghoy [31], (2018)	0.94 (Sensor data)
Xie [33], (2018)	0.89

**Table 4 sensors-23-00738-t004:** Quality assessment studies analysis regarding the categories.

Types of Data	First Author [Reference], (Year)	Device	Samples	Ripening Stages	Results	Total Score	Study Quality
Non-Augmented	Mendoza [43], (2004)	1	0	1	1	3	Fair
	Mazen [6], (2019a)	1	1	1	1	4	Strong
	Adebayo [18], (2017)	1	1	1	0	3	Fair
	Adebayo [17], (2016)	1	1	1	1	4	Strong
	Saranya [9], (2021)	1	1	1	0	3	Fair
	Ramadhan [19], (2019)	1	1	1	0	3	Fair
	Zhuang [20], (2019)	1	1	1	0	3	Fair
	Hou [22], (2015)	1	0	1	1	3	Fair
	Adebayo [35], (2017)	1	1	1	0	3	Fair
	Cho [10], 2021	1	1	1	0	3	Fair
	Zhang [23], (2018)	1	0	1	0	2	Weak
	Kipli [38], (2018)	0	1	1	1	3	Fair
	Mohamedon [41], (2021)	0	1	1	1	3	Fair
	Saragih [44], (2021)	0	1	1	1	3	Fair
	Zhu [24], (2021)	0	0	1	1	2	Weak
	Mueez [25], (2020)	1	1	1	0	3	Fair
	Maimunah [27], (2019)	1	0	1	0	2	Weak
	Zulkifli [4], (2019)	1	0	1	0	2	Weak
	Saadl [30], (2009)	1	0	1	1	3	Fair
	Xie [33], (2018)	1	0	0	1	2	Weak
	Mohapatra [5], (2017)	1	0	1	1	3	Fair
Highly Augmented	Sabilla [21], (2019)	0	1	1	1	3	Strong
	Zhu [26], (2020)	0	0	1	0	1	Weak
	Marimuthu [39], (2017)	0	1	1	0	2	Fair
	Vaviya [34], (2019)	0	0	1	1	2	Fair
Sensor-Based	Altaf [37], (2020)	1	0	1	1	3	Fair
	Suthagar [11], (2021)	0	1	1	0	2	Weak
	Taghoy [31], (2018)	1	0	1	0	2	Weak
	Sanaeifar [36], (2014)	1	1	1	1	4	Strong
	Chen [40], (2018)	1	0	1	0	2	Weak

## Data Availability

Not applicable.

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
