# Peer review of "Non-Destructive Banana Ripeness Detection Using Shallow and Deep Learning: A Systematic Review"

_sensors, 2023, doi:10.3390/s23020738_

Round 1
Reviewer 1 Report
The submitted article discusses the non-destructive testing of banana ripeness. For this purpose, modern deep or machine learning methods are used. This is an overview article, where the authors have conducted a detailed research and summarize individual aspects of research in this area. The research is done with high quality, the authors cite a sufficient number of literary sources. The issue is highly topical and can find application mainly in the food industry, but also in related industries. The main positives of this approach are its non-destructiveness, contactlessness, speed of analysis and other attributes. The whole process is based on the potential of computing technology, together with the implementation of neural networks, or deep and machine learning. The topic is treated clearly and comprehensibly in the article. the article has a clear structure. The authors declare the innovation of this approach, supporting the claims with relevant cited sources. In conclusion, I conclude that the presented issue is processed at a high professional level and therefore I recommend the article for publication in the given form.
Author Response
Thank you so much for your kind words, thoroughly reviewing and giving your valuable feedback in our review paper. I sincerely appreciate your effort for carefully reading our manuscript and consider our study to be published. We will be working further, in future, to improve the work we have done and make it more relevant with the real world scenarios.
Your appreciative feedback has been a driving force for us to do much better in future and we will try to keep up the momentum for our future work in this field.
Reviewer 2 Report
I want to congratulate the authors on this beautiful article. This paper is wonderfully articulated and has an interest in the academic point of view as well as the agricultural industry point of view. The authors have summarized a variety of techniques for the topic under investigation. Following are some important issues to be discussed and revisions must be submitted.
The ripeness detection must be addressed in consideration of - How to predict future fault with the collected raw data of the present moment? How to deal with the diversity between the data distributions of present and future moments? Traditional machine learning algorithms can only resolve classification or regression issues within the same data distributions.
What about the classification of faults using blind data (No labels)? How will you address this? Suggest a methodology by applying the trained model to classify blind datasets. You may refer to "Figure 8: Framework for classification of blind data" from the following article. DOI is https://doi.org/10.36001/ijphm.2020.v11i2.2929
Tree-based algorithms exhibit significant performance and are able to explain the reasoning behind decisions by ML model. Refer to the paper and suggest white box modeling using tree-based algorithms.
Elaborate on the possibility of misclassification of a healthy condition as faulty depending on the degree of fault? Suggest a methodology.
Elaborate on the possibility of misclassification of a faulty condition as healthy depending on the degree of fault. If the model is deployed in real-time and such a situation arises, how will you identify that the fruit is in the failed and showcased as healthy by your system? Suggest a methodology.
A review of some deep learning algorithms should be extended.
The conclusion is very poor and needs to be written meticulously. The future directions are so weak. Most of my comments may fall into a discussion of future directions.
All the best! Looking forward to seeing your revision soon!
Author Response
Thank you so much for thoroughly reviewing and giving your valuable feedback in our review paper. I sincerely appreciate your effort for carefully reading our manuscript. Please find the enclosed attachment of our response to cater your feedback.

Round 2
Reviewer 2 Report
The authors have addressed my comments.
Author Response
Thank you so much for thoroughly reviewing and giving your valuable feedback on our review paper. I sincerely appreciate your effort in carefully reading our manuscript and considering our study to be published.